# Flexible, Strong and Multifunctional Anf@Ag Nanocomposite Film for Human Physiology and Motion Monitoring

Haofan Long [1,†], Qing Li [1,†], Shulan Peng [1], Shiqiang Chen [1], Tonghua Zhang [1], Mingyuan Zhang [2], Minghua Li [2] and Lei Chen [1,2,*]

1   College of Sericulture, Textile and Biomass Sciences, Southwest University, Chongqing 400715, China; longhaofande163@163.com (H.L.); qingli@swu.edu.cn (Q.L.); pengshulan@luolai.com.cn (S.P.); csq123163@163.com (S.C.); zhtonghua@aliyun.com (T.Z.)
2   State Key Laboratory of Bio-Fibers and Eco-Textiles, Qingdao University, Qingdao 266071, China; zhangyuanming001@163.com (M.Z.); hw7812@163.com (M.L.)
*   Correspondence: raychen@swu.edu.cn
†   These authors contributed equally to this work.

**Abstract:** To expand the application range of flexible pressure sensors, endowing them with multi-function capabilities becomes extremely important. Herein, a flexible, strong and multifunctional nanocomposite film was prepared by introducing silver nanoparticles (Ag NPs) into aramid nanofiber (ANF) film using a simple two-step vacuum filtration method. When the Ag content was 27.6 vol%, the electrical resistance of the resulting ANF@Ag nanocomposite film was as low as 1.63 $\Omega/cm^2$, and the water contact angle of the nanocomposite film reached 153.9 $\pm$ 1°. Compared to the ANF film, the tensile strength of the nanocomposite film increased from 55 MPa to 66.3 MPa with an increase of 20.5%. After being applied to the human body, the nanocomposite film served as a pressure sensor that was able to recognize different stimuli for healthcare monitoring. Based on the advantages, it may become a potential candidate for electronic skin, intelligent wearable devices and medical detection equipment.

**Keywords:** aramid nanofiber; nanocomposites; mechanical properties; multifunction; pressure sensor

## 1. Introduction

Flexible sensors have the characteristics of softness and flexible compared with traditional sensors, which makes them more widespread in the fields of smart wearable devices. Flexible sensors are divided into flexible pressure sensors [1,2], flexible temperature sensors [2–6], flexible optical sensors [7–9], flexible humidity sensors [8,10–12] and so on. Among them, flexible pressure sensors have drawn much attention on account of their low energy consumption [13,14], high sensitivity [15], long-term stability [16,17], fast response time [18–20], low detection limit [21] and wide monitoring range [21,22] to sense weak changes and convert them into electrical signals.

In recent years, more and more materials with outstanding performance have been used to prepare flexible pressure sensors, such as polydimethylsiloxane [22,23], polyvinylidene fluoride [14,24], polyethylene [14] and polyamide [23,24]. In addition to the above polymer materials, aramid nanofiber (ANF) as a new building block, with excellent mechanical properties, thermal stability and chemical erosion resistance [25–28], is also an ideal candidate as a raw material for flexible pressure sensors. Recently, Tang et al. [29] used ANFs to reinforce the layer-by-layer assembled reduced graphene oxide (rGO)@polyurethane (PU) nancompositE. The obtained nanocomposite with excellent mechanical properties could be served as a flexible pressure sensor with remarkable sensitivity and repeatability. Wang et al. [28] prepared an aerogel composed of ANFs and MXene, which could work as a pressure sensor to monitor human biological signals under extreme conditions. However, most of the previous reports focus on enhancing the sensitivity and stability of the flexible

pressure sensors, and few researchers pay attention to the multifunction of the sensors, which greatly limits their application. For example, when being applied in the fields of medicine and health care, antibacterial properties are usually important for the pressure sensors; when being applied in intelligent wearable devices, superhydrophobicity can prolong the service lifetime of the pressure sensors. Thus, developing a multifunctional pressure sensor with antibacterial properties and superhydrophobicity is of great significance.

In this study, Ag nanoparticles- (Ag NPs) deposited ANF nanocomposite film (ANF@Ag) was prepared by a simple two-step method. Firstly, flexible and porous ANF film was obtained via vacuum filtration, followed by freeze-drying to fix the 3D network structure, which facilitated the loading of the Ag NPs. Secondly, a mixture of Ag NPs and polydimethylsiloxane (PDMS) was introduced into the ANF film via vacuum filtration to obtain the ANF@Ag nanocomposite film. This nanocomposite film, with great mechanical properties, superhydrophobicity and suitable antibacterial properties, can be used as a flexible pressure sensor. Based on the advantages, the sensor has good development prospects in electronic skin [30,31], intelligent wearable devices [32,33], medical detection equipment [34,35], health monitoring facility [33,35], human machine interfaces [36,37] and soft robotics [38].

## 2. Experimental

### 2.1. Materials

Acetone (>99.5%), potassium hydroxide (KOH, ≥85%), dimethyl sulfoxide (DMSO, >99%), sodium borohydride ($NaBH_4$), polyvinylpyrrolidone (PVP, $M_w \approx 1{,}300{,}000$), silver nitrate ($AgNO_3$) and polydimethylsiloxane (PDMS) were purchased from Chongqing Chuandong Chemical Group Co., Ltd. (Chongqing, Sichuang). Sodium chloride (NaCl, >99.5%) and sodium citrate ($Na_3C_6H_5O_7 \cdot 2H_2O$, >99%) were supplied by Chongqing Yuexiang Chemical Group Co., Ltd. (Chongqing, Sichuang). Tetrahydrofuran (THF) was provided by Chengdu Jinshan Chemical Reagent Co., Ltd (Chengdu, Sichuang).

### 2.2. Preparation of the Ag NPs

Subsequently, 2.5 mL of $AgNO_3$ ($2.5 \times 10^3$ mol/L) solution and 1 mL of PVP (1 mol/L) were placed in a round bottom flask and magnetically stirred in ice water. Then, 1.5 mL ($2.5 \times 10^3$ mol/L) of $Na_3C_6H_5O_7 \cdot 2H_2O$ solution and 0.3 mL (0.1 mol/L) of $NaBH_4$ solution were slowly added in the above solution. After the completion of the reaction, the mixture was centrifuged and then rinsed repeatedly with deionized water three times. Finally, Ag NPs were obtained under vacuum drying for 12 h.

### 2.3. Preparation of the ANF Dispersion

The ANF dispersion was prepared as reported by Kotov [38]. Briefly, 0.6 g of the aramid fibers was immersed in a solution containing 0.6 g of KOH and 200 mL of DMSO. After being magnetically stirred for one week at room temperature, a homogeneous ANF dispersion was obtained.

### 2.4. Preparation of the ANF@Ag Nanocomposite Film

The ANF@Ag nanocomposite film was prepared by a two-step method; a schematic illustration of the preparation processes of the nanocomposite film is depicted in Figure 1. Step I: 5, 10, 20, 40 and 80 mL of the ANFs dispersion (2.0 mg/mL) were poured onto a microporous filter membrane under vacuum, respectively, followed by freeze drying under −60 °C to fix the nanonetwork structure of the resulting ANF films. Step II: a Ag NPs/PDMS/THF mixture (the mass ratio of Ag NPs:PDMS:THF = 3:10:987) was introduced into the ANF film by vacuum filtration under room temperature (about 25 °C) to obtain a multifunctional ANF@Ag nanocomposite film with 6.9, 13.8, 20.7, 27.6 and 34.6 vol% of Ag NPs.

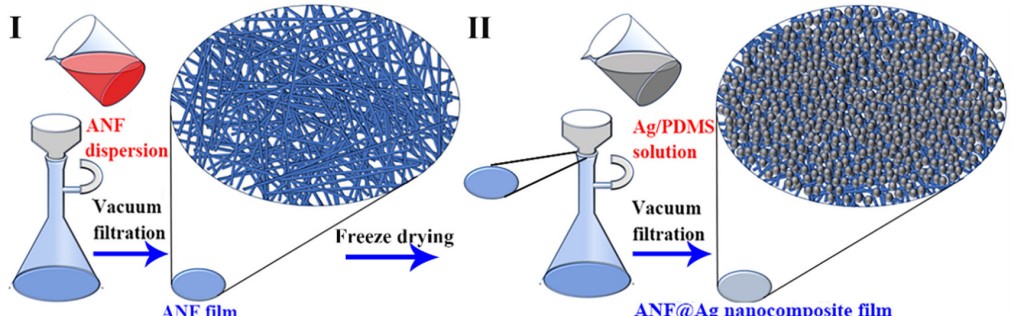

**Figure 1.** Schematic illustration of the preparation processes of the ANF@Ag nanocomposite film.

*2.5. Characterization and Measurement*

The morphology of the Ag NPs was observed by transmission electron microscopy (JEM-2100F, Tokyo, Japan). The particle size of the Ag NPs was analyzed via a nanoparticle analyzer (Zetasizer Nano 590, Malvern Pananalytical Ltd., Malvern, UK). The surface composition of the aramid fibers and ANFs was performed by Fourier transform infrared spectroscopy (Bruker Alpha, Bruker, Billerica, MA, USA). The morphology of the ANF film and ANF@Ag nanocomposite film was observed by scanning electron microscopy (JSM-6610, Jeol, Tokyo, Japan) equipped with an energy dispersive spectrometer (EDS). The crystal structures of the Ag NPs, ANF film and ANF@Ag nanocomposite film were measured by X-ray diffractometry (Smartlab 9 kW, Rigaku, Tokyo, Japan) using molybdenum target as the sourcE. Raman measurements of the ANF film and ANF@Ag nanocomposite film were characterized by a Raman spectrometer (Thermofisher DXR, Waltham, MA, USA) under an excitation light source of 532 nm.

The mechanical properties of the ANF film and ANF@Ag nanocomposite film were tested by an electronic universal testing machine (Autograph AGS-X, Shimadzu, Tokyo, Japan) with a gauge length of 10 mm at an extension rate of 5 mm/min. The thermal stabilities of the ANF film and ANF@Ag nanocomposite film were analyzed by a thermogravimetric analyzer (209 F3 Tarsus, Erich Netzsch GmbH, Selb, Bayern, Germany) in a nitrogen flow over a temperature range from 30 to 800 °C with a heating rate of 10 °C/min. The water contact angle of the ANF@Ag nanocomposite film was examined by a video optical contact angle meter (OCA15EC, Neurtek, Schaan, Liechtenstein). The infrared (IR) image was taken with IR thermography (HIKMICRO H11, Hangzhou, China). The antibacterial properties of the ANF@Ag nanocomposite film were evaluated in *E. coli* and *S. aureus* by a disc diffusion method. The electrical resistivity of the ANF@Ag nanocomposite film was performed by a four-point probe meter (DMR-1C, Wuhan, China). The electrochemical performance of the ANF@Ag nanocomposite film was tested by an electrochemical workstation (CHI660E, Corrtest Instruments, Wuhan, China).

**3. Results and Discussion**

*3.1. Preparation and Microstructure*

As shown in Figure 2a, the surface of aramid fibers was neat and smooth, and the color was light yellow with a diameter about 10 μm. After being stirred in KOH/DMSO solution for one week, a homogeneous and dark red ANFs dispersion could be obtained, and the ANFs had an average diameter of about 20 nm and length of several micrometers (Figure 2b). To detect changes in the surface composition between the aramid fibers and ANFs, FTIR measurements were performed (Figure 2c). FTIR spectrum of aramid fibers showed the stretching vibration peaks of the N-H bond at 3325 cm$^{-1}$ and C=O bond at 1651 cm$^{-1}$ in amides. The weak and sharp band at 1516 cm$^{-1}$ and the band at 1545 cm$^{-1}$ in the spectrum of the ANF film corresponded to the C=C skeletal vibration of the benzene ring in aramid macro-fibers. The peak around 1318 cm$^{-1}$ was induced by the Ph-N vibration, and the peak at 1018 cm$^{-1}$ was related to the in-plane C-H vibration [39]. FTIR spectroscopy of ANF film (Figure 2c) showed analogical characteristic peaks to that

of aramid fibers, indicating that nano-scale fibers retain their chemical structure despite the reduction in diameter [40]. Compared with the FTIR spectrum of the aramid fibers, there was no change in the position of the characteristic peaks, demonstrating that the ANFs maintained the basic structure of the aramid fibers.

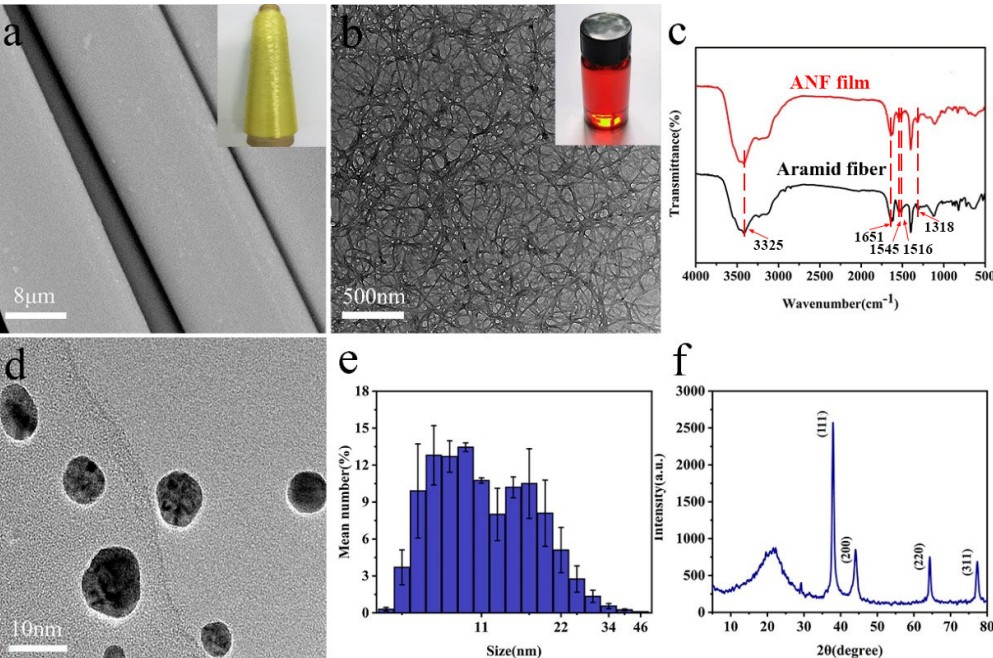

**Figure 2.** (**a**) SEM image and digital photograph (the inset) of the aramid fibers; (**b**) TEM image and digital photograph (the inset) of the ANF dispersion; (**c**) FTIR spectra of the aramid fiber and ANF film; (**d**) TEM image; (**e**) size distribution; and (**f**) XRD spectrum of the Ag NPs.

As shown in Figure 2d, the Ag NPs were spherical without agglomeration. The particle size was mainly concentrated in the range of 7 to 22 nm (Figure 2e), which was consistent with the TEM result (Figure 2d). To further prove the successful preparation of the Ag NPs, the crystal structure of the Ag NPs was characterized by XRD (Figure 2f). The diffraction peaks at 37.9°, 44.1°, 64.3° and 77.3° corresponded to the (111), (200), (220) and (311) planes of Ag NPs (card no: 34-0394 and 65-2871) [41].

Digital photographs, cross sections and surface morphologies of the ANF films prepared by different amounts of the ANF dispersion are shown in Figure 3. With the increase of the amounts of the ANF dispersion, the color of the ANF films gradually deepened (Figure 3a–e), and the thickness of the ANF films increased from 1.5 to 10.6 μm (Figure 3f–j). When the amounts of the ANF dispersion increased from 5 to 20 mL (Figure 3k–m), the ANF films showed a porous structure, but the average pore size of the films was different, decreasing from 658 nm to 224 nm. When the amounts of the ANF dispersion further increased (Figure 3n,o), the porous structure of the ANF films nearly disappeared. It has been reported that the porous structure with a uniform porosity is beneficial for the sensitivity of the sensor [42], and as the pore diameter of the ANF film prepared by 20 mL of ANF dispersion was closer to the diameter of the Ag NPs, the Ag NPs were easy to embed in the film without large agglomeration [21]. Therefore, 20 mL of the ANF dispersion was selected for follow-up experiments. It was noteworthy that the porous structure of the ANF film may collapse due to the surface tension in the solution during the ordinary drying process [26] (Figure S1). Hence, freeze drying was adopted to maintain the nano-scale morphologies of the ANF film in this study.

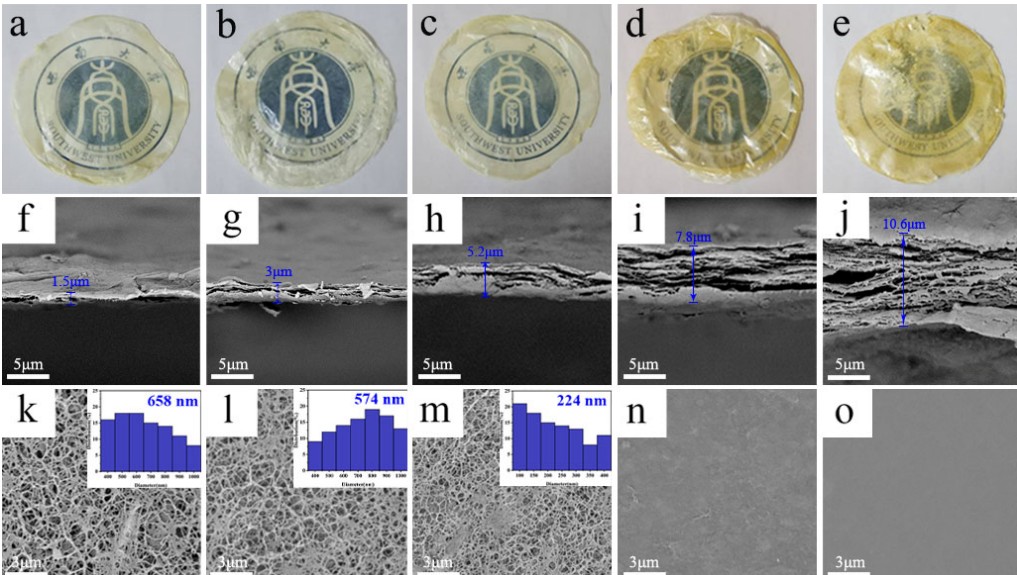

**Figure 3.** (**a–e**) Digital photographs, (**f–j**) cross section and (**k–o**) surface morphologies of the ANF films prepared by different amounts of the ANF dispersion: (**a,f,k**) 5 mL; (**b,g,l**) 10 mL; (**c,h,m**) 20 mL; (**d,i,b**) 40 mL and (**e,j,o**) 80 mL. The insets of Figure 3k–m show the pore size distribution of the ANF films prepared by 5, 10 and 20 mL of the ANF dispersion, respectively.

Compared with the yellowish color of the ANF film (Figure 3c), the color of the ANF@Ag nanocomposite film was dark brown (Figure 4a). The thickness of the ANF@Ag nanocomposite film (Figure 4b) was similar to that of the ANF film (Figure 3h), which were 5.8 μm and 5.2 μm, respectively. It could be observed that there were Ag NPs in the cross section of the nanocomposite film, suggesting the successful introduction of the Ag NPs.

Figure 4c–l displays the surface morphologies of the ANF@Ag nanocomposite films. After the introduction of the Ag NPs, the surface morphologies of the nanocomposite films changed a lot. As shown in Figure 4c–h, when the Ag content was low ($\leq$20.7 vol%), the Ag NPs were unevenly distributed on the surface of the film. When the Ag content reached 27.6 vol% (Figure 4i,j), the Ag NPs formed a uniform layer on the surface of the film. When the Ag content further increased to 34.6 vol% (Figure 4k,l), the Ag NPs aggregated on the surface of the film, indicating that there were no more binding sites for the Ag NPs.

The surface composition of the ANF film (Figure 5a) and ANF@Ag nanocomposite film (Figure 5b) was detected by EDS analysis. Besides C, O and N, Si and Ag appeared for the ANF@Ag nanocomposite film, further proving the successful introduction of the Ag NPs. As shown in Figure 5c, in the XRD pattern of ANF film, diffraction peaks appeared at 23° and 28.7°, corresponding to their (200) and (004) crystal planes, respectively. In the case of the ANF@Ag nanocomposite film, diffraction peaks appeared at 37.9°, 44°, 64.2° and 77.3°, which corresponded to the (111), (200), (220) and (311) planes of Ag NPs [41]. The ANF@Ag nanocomposite film had the peaks of Ag NPs as well as of ANF film. The XRD results revealed that the Ag NPs maintained the original crystalline domains after being introduced into the film. The Raman spectra of the ANF film and ANF@Ag nanocomposite film are presented in Figure 5d. The characteristic peaks of the C=C stretching vibrations of aromatic ring appeared at 1511 cm$^{-1}$ and 1610 cm$^{-1}$, and the C=O stretching vibrations appeared at 1650 cm$^{-1}$. There was a Ph-N vibration at 1321 cm$^{-1}$. The peaks at 1567 cm$^{-1}$ corresponded to the C-N stretching and coupled N–H in-plane deformation [38]. By contrast, the peak intensity of the ANF@Ag nanocomposite film increased dramatically due to the strong surface-enhanced Raman scattering (SERS) effect of the Ag NPs [42]. Since the smooth single crystal surface did not exhibit the SERS effect, it indicated that the Ag NPs formed a uniform and dense Ag layer with a certain roughness on the surface of the ANF film.

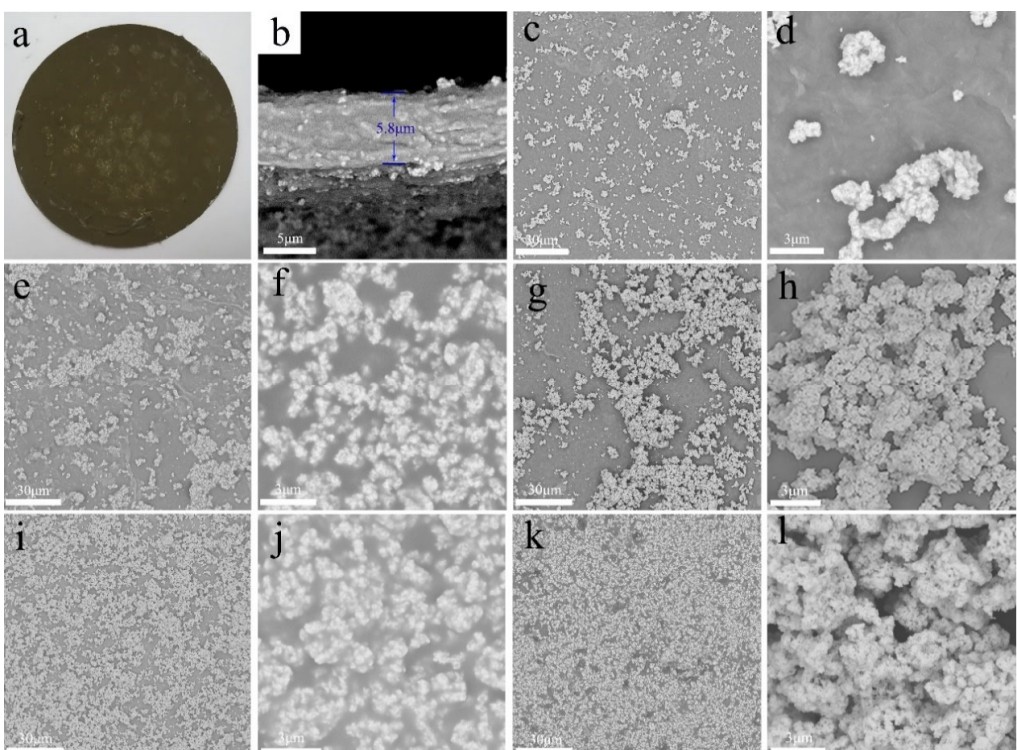

**Figure 4.** (**a**) Digital photograph and (**b**) cross section morphologies of the ANF@Ag nanocomposite film; surface morphologies of (**c–l**) the ANF@Ag nanocomposite films with different Ag content: (**c**,**d**) 6.9 vol%; (**e**,**f**) 13.8 vol%; (**g**,**h**) 20.7 vol%; (**i**,**j**) 27.6 vol% and (**k**,**l**) 34.6 vol%.

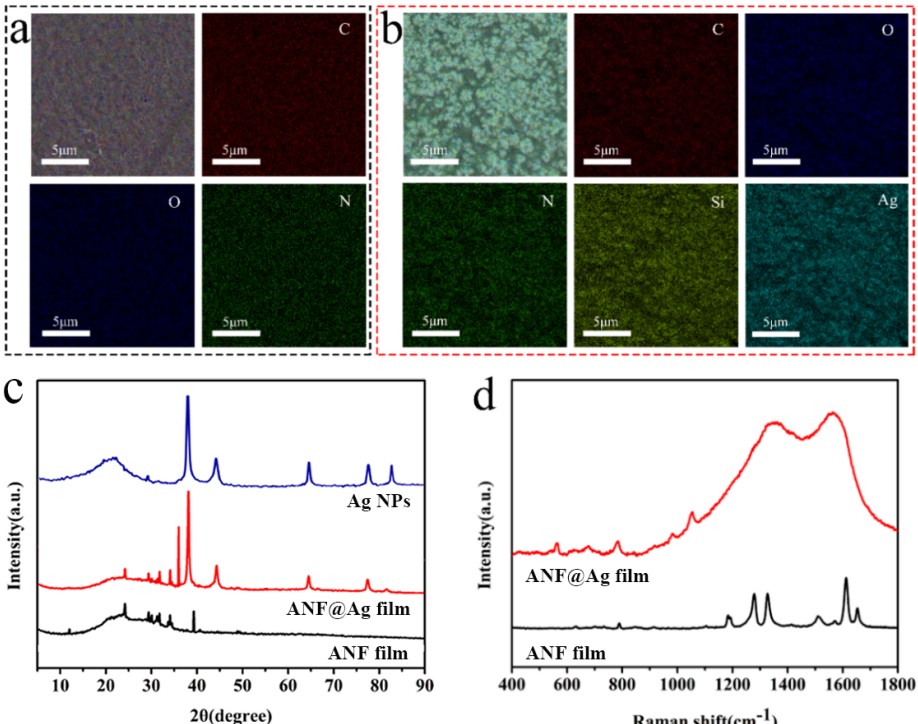

**Figure 5.** EDS mapping of the (**a**) ANF film and (**b**) ANF@Ag nanocomposite film; (**c**) XRD of the Ag NPs, ANF film and ANF@Ag nanocomposite film; and (**d**) Raman spectra of the ANF film and ANF@Ag nanocomposite film.

### 3.2. Physical Properties

The typical stress–strain curves of the ANF film and ANF@Ag nanocomposite films with different Ag contents wearere depicted in Figure 6a. The tensile strength of the ANF film was 55 MPa. When the Ag content was low (≤20.7 vol%), the tensile strength of the ANF@Ag nanocomposite films was lower than that of the ANF film. The tensile strengths of the ANF@Ag nanocomposite film with 27.6 vol% and 34.6 vol% of Ag content were 66.3 MPa and 67.9 MPa, respectively, which correspond to 20.5% and 23.5% increases in comparison with that of the ANF film. The changes of Young's modulus of the nanocomposite films showed the similar trend with the tensile strength (Figure S2). Figure S3 displays that the nanocomposite film (27.6 vol%) completely supported about a 205 g mass, implying its great mechanical properties. The optimization between the mechanical properties and electrical conductivity of the nanocomposite film could be achieved by adjusting the Ag content. The fracture processes of the ANF film and ANF@Ag nanocomposite films with low and high Ag contents were analyzed, as demonstrated in Figure 6b. According to the weak link theory, the maximum tensile strength of the nanocomposite film depended on the weakest fracture path, and Ag NPs could have had two opposite effects on the tensile strength of the nanocomposite film according to their distribution state [26]. One is the stress concentration caused by the uneven distribution of the Ag NPs in the nanocomposite film with a low Ag content (≤20.7 vol%), resulting in the fracture in the area without Ag NPs. In this case, the Ag NPs had a weakening effect on the tensile strength of the nanocomposite film. The other was that the uniformly distributed Ag NPs acted as obstacles to crack propagation and had an enhancing effect on the tensile strength of the nanocomposite film with a high Ag content (≥27.6 vol%).

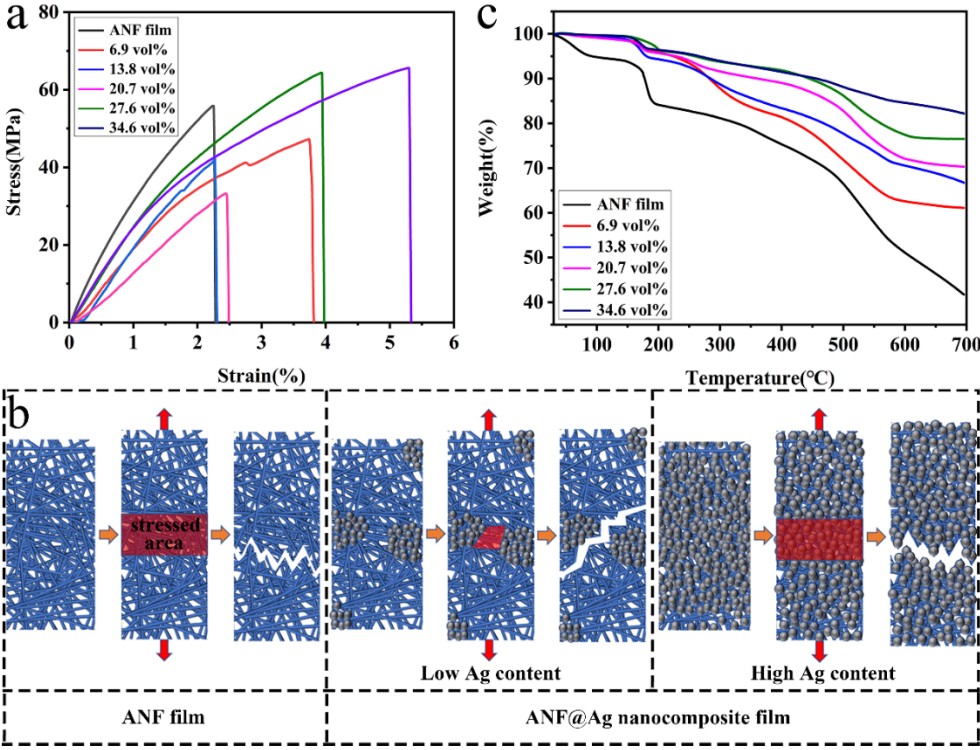

**Figure 6.** (**a**) Stress–strain curves of the ANF film and ANF@Ag nanocomposite films with different Ag contents; (**b**) schematic illustration of the fracture processes of the ANF film and ANF@Ag nanocomposite films; and (**c**) TG analysis of the ANF film and ANF@Ag nanocomposite films with different Ag content.

The thermal stabilities of the ANF film and ANF@Ag nanocomposite films with different Ag contents were analyzed by TG. The ANF film presented two main mass loss processes (Figure 6c). The first mass loss temperature was about 100 °C, which was due to

the evaporation of water and the decomposition of molecular chains with a large number of polar end groups. The second mass loss temperature was around 170 °C, which was mainly due to the damage of the aramid crystal structurE. The onset decomposition temperatures ($T_{5\%}$) of the ANF film and ANF@Ag nanocomposite films with different Ag contents were 92.4, 185.1, 223.1, 227.9, 258.5 and 265.3 °C, respectively. The results show that the thermal stability of the ANF@Ag nanocomposite film was better than that of the ANF film because silver has a thermal dissolution temperature of more than 900 °C. Thus, the thermal stability of the ANF@Ag nanocomposite films increases obviously with the increasing of Ag content.

### 3.3. Electrical Conductivity, Surface Wettability and Antibacterial Properties

The effects of the Ag content on the electrical conductivity of the ANF@Ag nanocomposite films were investigated, as shown in Figure 7a. With the increase of the Ag content, the sheet resistance of the nanocomposite film decreased gradually. When the Ag content was 27.6 vol%, the sheet resistance was as low as 1.63 $\Omega/cm^2$. Combined with the SEM results (Figure 4i,j), it indicated that the Ag content had reached saturation on the surface of the film. When the Ag content further increased to 34.6 vol%, it seemed that the sheet resistance did not change anymorE. Taking the production cost and environmental protection into consideration, 27.6 vol% of the Ag content was selected for follow-up experiments. The ANF@Ag nanocomposite film (27.6 vol%) was assembled into a current circuit (Figure S4). When the circuit was connected, the bulb was lit with a consistent brightness. To further study the adhesion strength between the Ag NPs and the ANF matrix, repeated adhesion-peeling (Figure 7b) and bending–unbending cycle (Figure 7c) tests were performed on the ANF@Ag nanocomposite film. After 100 cycles of adhesion-peeling and 1000 cycles of bending–unbending, the sheet resistance increased by only 7.9% and 6%, respectively, demonstrating that the Ag NPs were firmly bonded to the film via PDMS. The nanocomposite film exhibited outstanding conductive stability, which lays a solid foundation for its development in flexible pressure sensors.

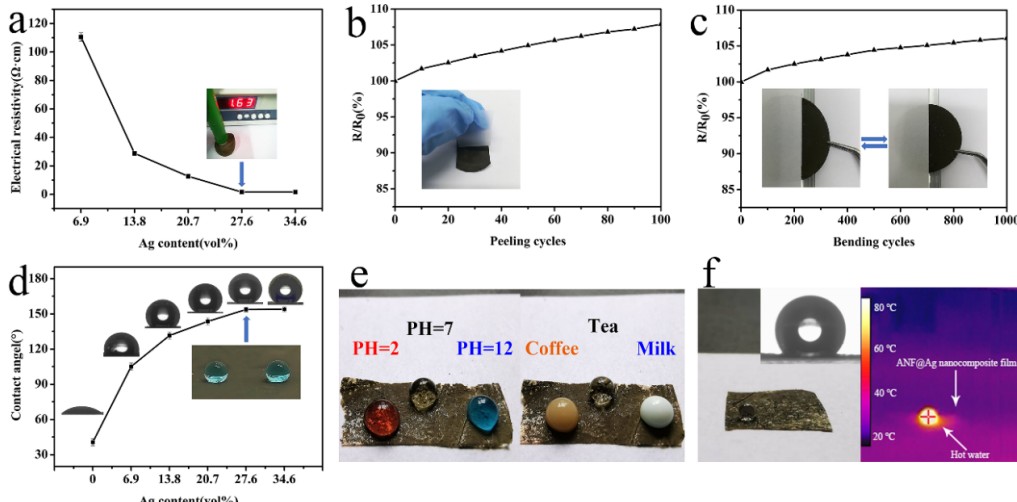

**Figure 7.** (**a**) Effects of Ag content on the electrical conductivity of the ANF@Ag nanocomposite film; (**b**) effects of adhesion-peeling cycles (the inset shows that an adhesive tape is used to adhere and peel off from the nanocomposite film) and (**c**) bending–unbending cycles (the inset shows that the nanocomposite film wraps around a glass rod) on the changes of the sheet resistance of the ANF@Ag nanocomposite film; (**d**) effects of Ag content on the contact angle of the ANF@Ag nanocomposite film (the inset shows water droplets on the surface of the film); (**e**) digital photographs of solutions with pHs of 2, 7 and 12 and three common drinks including coffee, tea and milk on the surface of the ANF@Ag nanocomposite film; and (**f**) digital photograph and IR image of a hot water droplet on the surface of the ANF@Ag nanocomposite film.

As pressure sensors are usually applied in environments with a high humidity (e.g., sweat and rain), it is significant to endow the sensors with superhydrophobicity. Figure 7d shows the contact angles of the ANF@Ag nanocomposite films with different Ag contents. Obviously, the ANF film with a contact angle of 43.8 ± 1.5° was hydrophilic and would absorb water easily. After the introduction of the Ag NPs (6.9 vol%), the surface of the film became hydrophobic with a contact angle of 105.1 ± 1.5°. When the Ag content increased to 27.6 vol%, the surface of the film exhibited superhydrophobicity with a contact angle of 153.9 ± 1°. With a hierarchical roughness derived from Ag NPs, like the mastoid structure of a lotus leaf (Figure S5), the micro/nano hierarchical structures increase surface roughness dramatically so that air can be trapped in the aperture of Ag NPs and nanofibers, even below water droplets, and thus reduce the liquid–solid contact and cause superhydrophobicity. Besides the low surface energy derived from PDMS, this was also a key factor for constructing the superhydrophobic surfacE. When acid (red, pH = 2), pure water (transparent, pH = 7), alkaline (blue, pH = 12) and three common drinks including coffee, tea and milk were dripped on the surface of the ANF@Ag nanocomposite film (27.6 vol%), spherical droplets steadily stood on it, demonstrating good corrosion resistance (Figure 7e). Interestingly, the ANF@Ag nanocomposite film revealed hot water repellency (Figure 7f). The IR image of a hot water droplet (70 °C) on the surface of the ANF@Ag nanocomposite film showed a contact angle of 151.3 ± 1.1°, which was similar with that of a room-temperature water droplet on the surface of the nanocomposite film. This phenomenon was ascribed to the fact that the layer of the Ag NPs maintained integrity in hot water.

Wearable electrical devices are usually directly contacted with human skin; hence, it is necessary to endow them with antibacterial properties for providing a germ-free environment and increasing their service lifetimE. The antibacterial properties of the ANF@Ag nanocomposite film were measured by a disc diffusion *E. coli* and *S. aureus* (Figure 8). An inhibition zone could be observed around the antibiotic blank sample, which indicated that the bacteria were successfully cultured. As expected, there was no inhibition zone around the ANF film, indicating that it did not have antibacterial properties. In contrast, it could be clearly seen that there were inhibition zones with diameters of 16.5 and 18 mm around the ANF@Ag nanocomposite film against *S. aureus* (ATCC8739) and *E. coli* (ATCC6538), respectively. The Ag NPs could interact with the components of the bacterial cell membrane, increase the permeability of the cell membrane, and then inhibit the growth of bacteria [43,44] (Figure S6a). Additionally, the Ag NPs could interact with the sulfhydryl groups of the L-Cysteine residue of the bacterial protein, thereby inactivating the enzyme [45,46] (Figure S6b).

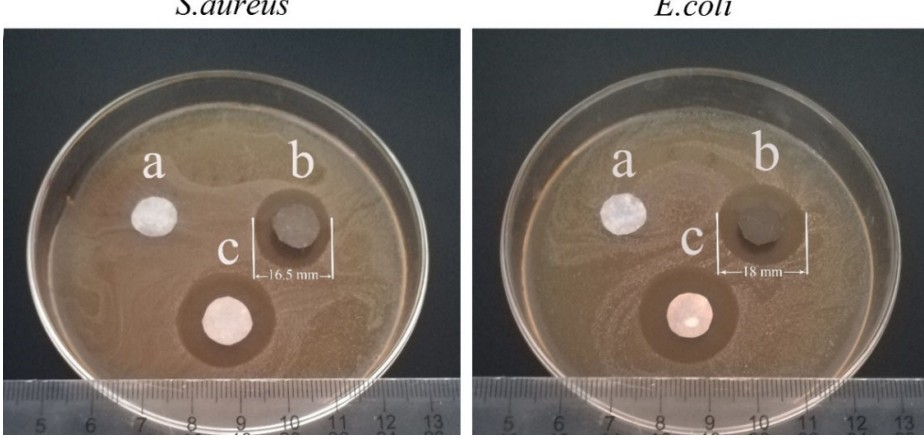

**Figure 8.** Digital photographs of the inhibition zone formed in antibacterial test of the (**a**) ANF film, (**b**) ANF@Ag nanocomposite film and (**c**) antibiotic blank sample (treated by tetracycline) against *E. coli* and *S. aureus*.

*3.4. Sensing Performance*

The schematic of the sensing mechanism of the ANF@Ag nanocomposite film as a pressure sensor is shown in Figure S7. When the sensor was subjected to external force, the distance and contact area of the Ag NPs changed to a certain extent, which made the contact resistance of the conductive pathway change accordingly. Finally, a sensing conversion from the external force to the electrical signal was achieved.

Good sensing durability and stability are critical for a pressure sensors and have been investigated by detecting the current changes under loading–unloading external forces (Figure 9a). After being tested for 3000 s, the sensing signal was constant, which directly proved that the sensor was durable and stablE. To demonstrate the feasibility of the sensor in monitoring human body movement and physiological signals, the sensor was attached to different human body parts. Clearly, the curve showed a stable variation of current caused by the pressing of a finger (Figure 9b). To further explore whether the sensor can monitor multiple parts of the human body, the sensor was employed to monitor the bending of the knee (Figure 9c) and wrist (Figure 9d). The curve demonstrated great repeatability, and small peaks caused by shakes revealed that the sensor was able to detect tiny movements with a high sensitivity [47]. Furthermore, the sensor was attached to the abdomen (Figure 9e) and throat (Figure 9f) of the volunteer to monitor physiological signals. Obviously, the curve of breathing showed a regular respiratory rhythm of about 2.5 s per round. The curve of speaking presented great fluctuations when the volunteer pronounced with a stress on "O". Both results revealed the great performance of the sensor in catching physiological signals.

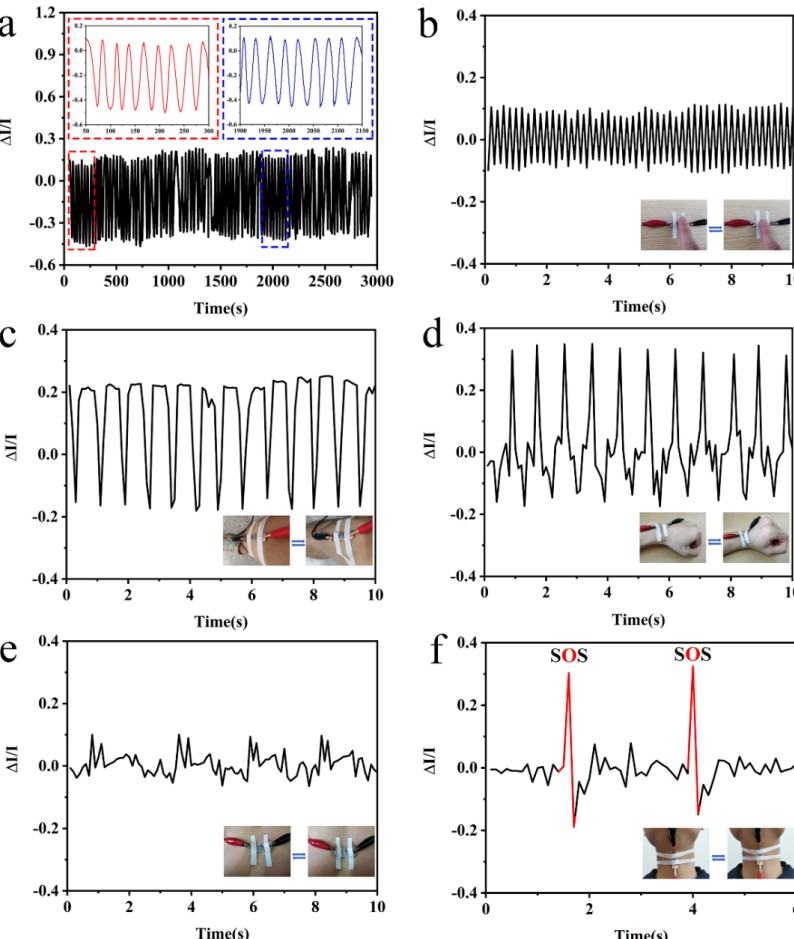

**Figure 9.** (**a**) Durability and stability testing results of the pressure sensor; (**b**–**f**) changes of sensing signals when the pressure sensor was attached to the human body caused by mechanical stimuli of (**b**) pressing of a finger, bending of the (**c**) knee, (**d**) wrist, (**e**) abdomen and (**f**) throat.

## 4. Conclusions

In conclusion, a flexible, strong and multifunctional ANF@Ag nanocomposite film was prepared by introducing Ag NPs into porous ANF film using a simple two-step vacuum filtration method. The nanocomposite film showed great mechanical properties and thermal stability. When the Ag content was 27.6 vol%, the nanocomposite film had a thickness of $5.8 \pm 0.5$ μm, which makes it light weight with a good flexibility. At the same time, the addition of Ag NPs endowed it with relatively high conductivity and outstanding conductive stability. In addition, the nanocomposite film was superhydrophobic, with good acid, alkaline and hot water resistance, and had suitable antibacterial properties against *E. coli* and *S. aureus*. When the nanocomposite film was used as a pressure sensor, the sensor with good sensing durability and stability could accurately recognize different physiological and motion stimuli of the human body. The results demonstrated that the combination of the ANFs and Ag NPs provide a feasible pathway to prepare multifunctional films used for wearable electronic sensors.

**Supplementary Materials:** The following supporting information can be downloaded at: https://www.mdpi.com/article/10.3390/pr10050961/s1, Figure S1. Surface morphologies of the ANF film prepared by ordinary drying process at (**a**) low and (**b**) high magnification. Figure S2. Young's modulus of the ANF film and ANF@Ag nanocomposite films with different Ag content. Figure S3. Digital photograph of demonstrating the ability of the ANF@Ag nanocomposite film (27.6 vol%) completely support ca. 205 g mass. Figure S4. Digital Photographs of the ANF@Ag nanocomposite film (27.6 vol%) connected to a circuit with a LED bulb. Figure S5. (**a**) Mastoid structure of lotus leaf and (**b**) Hierarchical structure of the ANF@Ag nano-composite film. Figure S6. Schematic illustration of the antibacterial mechanism of the Ag NPs. Figure S7. Schematic illustration of the ANF@Ag pressure sensor in response to different exter-nal mechanical stimuli.

**Author Contributions:** Data analysis, H.L. and S.C.; funding acquisition, L.C.; experimental, H.L. and T.Z.; investigation, S.P. and M.Z.; resources: M.L.; writing—original draft, H.L.; writing—review and editing, L.C and Q.L. All authors have read and agreed to the published version of the manuscript.

**Funding:** This work was financially supported by the National Natural Science Foundation of China (Grant No. 52003227), the State Key Laboratory of Bio-Fibers and Eco-Textiles (Qingdao University, Grant No. K2019-05), the Science and Technology Research Program of Chongqing Municipal Education Commission (Grant No. KJQN202100224) and the Venture & Innovation Support Program for Chongqing Overseas Returnees (Grant No. cx2020062).

**Informed Consent Statement:** All subjects gave their informed consent for inclusion before they participated in the study. The protocol was approved by the Chongqing Municipal Education Commission of KJQN202100224.

**Conflicts of Interest:** The authors declare no conflict of interest.

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
