# Peer review of "Flexible, Strong and Multifunctional Anf@Ag Nanocomposite Film for Human Physiology and Motion Monitoring"

_processes, doi:10.3390/pr10050961_

Round 1
Reviewer 1 Report
I have read "Flexible, strong and multifunctional ANF@Ag nanocomposite film for human physiology and motion monitoring" . I cannot, therefore, recommend this paper for publication in Processes in the present form. Changes need to be made to the manuscript before acceptance.
- In the Experimental part add the temperature of vacuum drying
- It is not possible to observe the changes in the FT-IR to identify the peaks in the spectrum.
- Space group and cell parameters of the prepared materials should be included.
- More details of the experimental part of the antibacterial properties type of agar, the bacteria are ATCC or isolated
- Mention in the conclusions the thickness of the film of Ag content 27.6 vol %
- Ultrasound is used in the dispersion of AFN ??
- Which antibiotic is used add at the bottom of the figure
- further discussion of surface wettability properties, add references
- Why perform antibacterial properties?
- How does thickness affect conductivity properties?
Reviewer 2 Report
Q1- Page 2, line 81: Please explain why DMSO used for the preparation of the ANF dispersion.
Q2-Page 3, line 127-132: FTIR results have not been sufficiently discussed. Indicate what changes were observed as a result of interaction with Ag nanoparticles. Support your results with literature. Indicate the main differences between FTIR spectra of the aramid fiber and ANF film.
Q3-Page 4, line 147-149: ANF films show porosity when ANF dispersion amounts increase from 5 to 20 mL, but the average pore size of the films decreases. Please explain the reason clearly.
Q4-Page 6, Figure 4 and corresponding discussion: Explain the difference in the surface morphologies of nanocomposite films after the introduction of Ag NPs.
Q5-Page 7, Figure 5 and corresponding discussion for XRD: Revise your XRD results by adding the XRD curve given for Ag nanoparticles in Figure 2f to the XRD graph in Figure 5c.
Q6-Page 7-8: At low Ag content, the tensile strength of ANF@Ag nanocomposite films is lower than that of ANF film. The improvement in tensile strength of ANF@Ag nanocomposite films with 27.6% vol% and 34.6% vol% Ag content is also not high, how do you explain this? What are the improvements in similar systems in the literature? For example, what is the Solvent effect used during preparation?
Q7-Page 8, line 231-240: TG results are also not sufficiently discussed. In how many stages does the decomposition of the films occur, which groups are degraded at each stage, how the amount of residue changes, please explain in detail.
Q8-Page 9, line 278-281, Figure 7e: Explain the reason for the changes in Ag nanocomposite film color when three common beverages, including acid (red, pH = 2), pure 278 water (clear, pH = 7), alkaline (blue, pH = 12) and 279 coffee, tea and milk, are dropped onto the ANF® surface.
Round 2
Reviewer 1 Report
the authors heeded all the recommendations that I made to them, therefore I agree that the manuscript be published
Reviewer 2 Report
The authors revised the article with adequate answers to the questions I asked. It is suitable for publication as it is.